# Crop Response to Combined Availability of Soil Water and Its Salinity Level: Theory, Experiments and Validation on Golf Courses

Jiftah Ben-Asher [1], Jose Beltrao [2,3], Gulom Bekmirzaev [4,*] and Thomas Panagopoulos [2]

1  French Associates Institute for Agriculture and Biotechnology of Dryland, The Jacob Blaustein Institutes for Desert Research, Sede Boqer Campus, Ben Gurion University of the Negev, Beer Sheva 84105, Israel; benasher@bgu.ac.il
2  Faculty of Science and Technology, Campus de Gambelas, Universidade do Algarve, 8005 Faro, Portugal; jbeltrao@ualg.pt (J.B.); tpanago@ualg.pt (T.P.)
3  Centro de Investigação Professor Doutor Joaquim Veríssimo Serrão, Casa de Portugal e de Camões, 2005 Santarém, Portugal
4  Department of Irrigation and Melioration, Tashkent Institute of Irrigation and Agricultural Mechanization Engineers, Kori-Niyoziy Str. 39, Tashkent 100000, Uzbekistan
*  Correspondence: gulombek@gmail.com; Tel.: +998-(97)-4608978

**Abstract:** The phenomenological expression showing crop yield to be directly dependent on water deficiency, under saline conditions, has encouraged a continued focus on salinity as a viable approach to increase crop yields. This work reassesses crop response to availability of saline soil water ASW in two stages (A) Develop a simple approach suggesting that permanent wilting point (WP) increases under high saline soil water tension and relative yield of Lettuce (*Lactuca sativa* L., var longifolia Lam., cv. Nevada) and maize (*Zea mays* L., cv. Jubilee sweet) decrease. (B) Using a deterministic numerical soil water model to validate the theory on Bermuda grass of golf courses. The experimental plots were established in the North Negev, Israel (Sweet corn) and the Algarve, Portugal (Lettuce and Bermuda grass covering the golf courses). Sprinkler irrigation and line source techniques were used for water application, creating a saline gradient under a precise irrigation water distribution. Two salinity empirical models were tested (Mass and Hoffman MH and van Genuchten–Gupta vGG). Their empirical models were modified and instead of soil electrical conductivity of irrigation water ($EC_e$) we used wilting point (WP) and RASW to follow the changes in relative yield. The validation was conducted with theoretical soil plant atmosphere water (SPAW) to predict the results on golf courses. It is concluded that an alternative S-shaped response model provides better fit to our experimental data sets. Modified MH model ($Yr = Y/Ymax = a * (ASW–threshold's constant)$) revealed that a single dimensionless curve could be used to express yield—salinity interference when represented by varying ASW. The vGG model: vGG can represent salt tolerance of most crops, by using varying wilting point of average root zone salinity, at which the yield has declined by 50%. The abscissa of both models was based on WP rather than the standard soil electrical conductivity ($EC_w$). The correlation between the experimental data and WP or relative available soil water (RASW) was acceptable and, therefore, their usefulness for prediction of relative yield is acceptable as well. The objectives of this study were: 1. To develop a simple model describing the effect of salinity through soil water availability on crop production; 2. To replace the standard varying soil electrical conductivity $EC_e$ used by MH and vGG models by two soil parameters (at wilting point- $\theta_{wp}$ and at field capacity $\theta_{fc}$) in order to describe the relationship between them and relative yield. 3. Validate the new model with respect to independent salinity on Golf courses and a mathematical deterministic model.

**Keywords:** relative yield; SPAW numerical model; $EC_e$ and saline soil water $EC_w$; sprinkler irrigation

## 1. Introduction

Salinity of soil and irrigation water is affecting several agronomic and environmental aspects. Soil salinization is one of the major soil degradation threats occurring in large areas of the word. It can be observed in numerous vital ecological and non-ecological soil functions. It reduces biomass production and hence due to reduced photosynthesis it is associated with decline of $CO_2$ sequestration. Thus, three major losses can be related to salinity. Two of them can be classified as agronomic losses (Yield reduction and indirectly the increased negative ecological input on the expenses of $CO_2$ sequestration.) The third is soil degradation that affect both agronomic and ecological functions. In this paper we proposed a new model in which the effect of salinization on soil is given through the reduction of water availability to plant due to increased soil water tension under saline conditions and the difference between field capacity and wilting point and yield cutback.

Linear relationships between transpiration and soil water content were analyzed by Sinclair [1]. In his analysis, the model of de Wit [2] was thoroughly discussed. Similar models were published later, supported by the results of de Wit [2–4]. However, one of Sinclair's [1] conclusions was that "transpiration efficiency" is not as critical to crop production as soil water content. Linear relationships between water use and yield have been modelled for various crops and climates under conditions of water deficit [5–7] and conditions of salt stress [8–10].

The salinity condition in the root zone hinders moisture extraction from soil by plants, because of osmotic potential development in soil water, due to the presence of salts, which ultimately decreases transpiration of plants, and thereby affects crop yield [11–13]. If salts are not properly managed in a timely fashion, they can pose a serious risk for irrigated crops [14–16] these findings could help the stakeholders of irrigated crops in adopting strategies leading to the reduction in the salinity in the crop root zone [17–19]. The effect of salinity on crop yield has been modeled with a piece-wise linear response model MH [20]. An inverted logistic exponential equation for quantifying crop salt tolerance was presented by van Genuchten and Hoffman [21], showing how salt tolerance data can be by coupling an appropriate salt balance model with a least-squares optimization method. Later, van Genuchten and Gupta [22] conducted a thorough study on this model that was "crowned" and widely used with the initials vGG.

The MH model is the most common representation of salt tolerance, describing crop salt tolerance by a threshold salinity parameter below which yield is not affected, and a slope parameter describing the decline in relative yield when salinity is beyond the threshold.

From an agronomic point of view, vGG model is more successful in farm lands than the MH model, but it has been less widely applied, as the parameters do not have the same intuitive appeal as the parameters of the MH threshold model [23]. He noted that the threshold value proposed by MH model is lower as yield is higher, and at very high yield the threshold is zero. Also, at low yields, the threshold values are not representative. It should be further emphasized that these quoted studies did not test the combined effect of changes in soil water availability and salinity on yield.

For non-saline soils, the conventional concept to define the availability of soil water [24–26], assuming that the water readily available to plants is the difference between the soil water content at field capacity ($\theta_{fc}$) and permanent wilting point ($\theta_{wp}$) is still used. Notice that high frequency irrigation takes saturated water content as the highest limit for water availability rather than field capacity ($\theta_{fc}$). However, the proposed model is focused on wilting point and not on the upper limit. In most cases, when calculating soil water availability (SWA), the effect of salinity on wilting point is neglected. Nevertheless, the higher the effect of salinity on wilting point is neglected. Nevertheless, the higher concentration of soil solution, the larger the soil water content at wilting point, and the smaller the availability of soil water [27].

The objectives of this study were:

1. To develop a simple model describing the effect of salinity through soil water availability on crop production.
2. To replace the standard varying soil electrical conductivity $EC_e$ used by MH and vGG models by two soil parameters ($\theta_{wp}$ and $\theta_{fc}$) in order to describe the relationship between them and relative yield.
3. Validate the new model with respect to independent salinity stady on Golf courses and a mathematical deterministic model.

## 2. Theory

### 2.1. Combining Salinity, Environment and Water Stress

For non-saline soils, the available soil water capacity ASW, following the concept of Veihemeyer and Hendrickson [24], was the range of available water that can be stored in soil and available for growing crops. It was assumed, by the same authors, that the ASW to plant ($\theta_{ASW}$) is given by:

$$\theta_{ASW} = \theta_{fc} - \theta_{wp} \tag{1}$$

where: $\theta_{fc}$ and $\theta_{wp}$ are the volumetric soil water content at field capacity and at wilting point respectively (%).

The water content at wilting point is a dynamic soil parameter. It is affected by salinity that is expressed by electrical conductivity (EC) or soil water tension. The plant can use less water at high EC than at low EC. In other words, $\theta_{wp}$ in Equation (2) remains high. Therefore, RASW and SWA under high EC are smaller than under low EC. The dynamic nature of $\theta_{wp}$ can be given by the factor $\Delta$ that describes the fraction of change in $\theta_{wp}$ by:

$$\theta_{wp\,(s)} = \theta_{wp} + \Delta\theta_{wp}; 0 \leq \Delta \leq 1 \tag{2}$$

The factor $\Delta$ is a dimensionless factor affected by several environmental conditions, among which only salinity can be controlled by leaching. It is large for high soil salinity, shallow rooted plants and high evaporative ability of the atmosphere. Under these conditions, high frequency irrigation should be applied to satisfy crop's water requirements, especially at high evaporative conditions. The volume (Vs) of available soil water (ASW) for transpiration ($m^3$) under saline conditions given by:

$$Vs = A * Z * \{\theta_{fc} - [\theta_{wp}\,(1 + \Delta)]\} \tag{3}$$

where: Z is the depth of the root zone (m) and A is the evaporating surface area ($m^2$).

### 2.2. Models for Crop's Yield and Salinity

Equation (3) relates depletion in water supply due to increased salinity and associated wilting points, rather than increased soil water tension and electrical conductivity. For many reasons it is more convenient to collect soil moister data than soil water tension or soil salinity data.

This model Equation (3) is supported by a new study conducted by Sinclair [1]. The study suggests that available soil water is more effectively used through the growing season than improved transpiration efficiency, which is controlled by partial stomatal close.

The findings of Sinclair are based on mechanistic analysis while many other studies de Wit [2] were based on experimental studies focusing on transpiration efficiency. Normalizing the results of yield and transpiration with respect to their maximum values was an important approach to improve the understanding of the processes. Passioura [28], for example, suggested a linear model, that after simple arithmetic arrangements in Equation (3) and converting harvest index to total above ground dry matter the relative yield can be written as a function of available soil water content under saline conditions. Equation (4) was inspired by the MH models and Sinclair model, and it is a modified model accordingly to soil water terminology, but remained with the mathematical structure of MH as follows:

$$Y_r = Y/Y_{max} = a * (ASW - Threshold's\ constant) \tag{4}$$

where: $Y_r$ = relative yield (dimensionless); ASW is the available soil water (%); "a" is the increase in $Y_r$ per unit increase in ASW (%). Threshold's constant: for ASW less than this parameter $Y_r \approx 0$ (%).

Linear relationships between water use and yield have been modeled by many scientists, especially under conditions of water deficit, due to the presence of salts [2–6,8,9]. This relationship of the crop yield response to soil water availability, under saline conditions or soil water potential, is also given by the vGG model (Equation (5)).

The difference between MH and vGG models is the continuity nature of vGG and the discontinuity of MH model. The last one contains a threshold (breakpoint), beyond which yield starts to decline until crop cannot take up its water requirement due to high soil EC [29], its associated water potential and reduced water depth at $\theta_{wp} (1 + \Delta)$.

To avoid some of the uniqueness problems with the MH threshold-slope model, we included alternative response functions that would give a more accurate description than the MH model. We selected Equation (5) which is a smooth, sigmoidal function of van Genuchten and Gupta [22] for further analysis, as follows:

$$Y_r = \frac{Y}{Y_0} = 100 \frac{1}{\left[1 + \left(\frac{\theta_{wp}}{\theta_{wp50}}\right)^p\right]} \qquad (5)$$

Having the yield changing with $\theta_{wp}$ and $Y_0$ maximal yield, the ratio $Y/Y_0 = 100$. That is, Equation (5) is a model that contains only two unknown parameters: the maximum yield ($Y_0$) and wilting point (affected by the soil salinity) at which the yield is declined to 50% of its maximum value ($\theta_{wp50}$). This unknown parameter can be extracted from the graph of Yr vs. the wilting point. Thus, the only unknown parameter left in Equation (5) is "p", a parameter that determines the steepness of the graph $Y_r$ vs. $\theta_{wp}$. The larger is the "p" the steeper is the slope of yield decline. Notice that salinities and yields in Equation (5) are normalized. Hence, they lead to a dimensionless plot in terms of two scaled variables (relative yield $Y/Y_m$ and relative wilting point $\theta_{wp}/\theta_{wp50}$) and one coefficient "p", reflecting the steepness of the curve.

In agricultural practice, the MH model is easier to interpret than the vGG, but mathematically, the vGG model is preferable because it resolves the MH model's discontinuity.

### 3. Materials and Methods

The horticultural crops selected for the study were sweet grain corn (*Zea mays* L., cv. Jubilee sweet) and summer lettuce (*Lactuca sativa* L., var longifolia Lam. cv. Nevada). The experimental sweet corn plots were established in the Ramat Negev Agro-research Center, Ashalim, North Negev, Israel (Lat. 31°05′00″ N, Long. 34°41′03″ E, Alt. 305 m). On the other hand, Lettuce experiments were carried out in Campus de Gambelas, University of Algarve, Faro, South Portugal (Lat. 37°02′35″ N, Long. 7°58′16″ E, Alt. 10 m). Sowing space for sweet corn was 0.5 m and its depth were about 0.02–0.03 m, being plant population density about 8 plants per $m^2$ [29]. On the other hand, planting space for the lettuce was 0.4 m × 0.3 m [30].

Sprinkler irrigation and line source techniques were used for water application, creating a saline gradient with a minimal experimental area [31–33].

The experiment was a modification of the single line source concept of Hanks et al. [31]. Original salinity was 7.0 $dSm^{-1}$. Samples of water applied to the plot were collected during each irrigation by funnels attached to collection bottles. Total water amount collectd was 202 mm and the cristiansen uniformity was 0.95. Ec decreased linearly awaiy from the saline line (Figure 1).

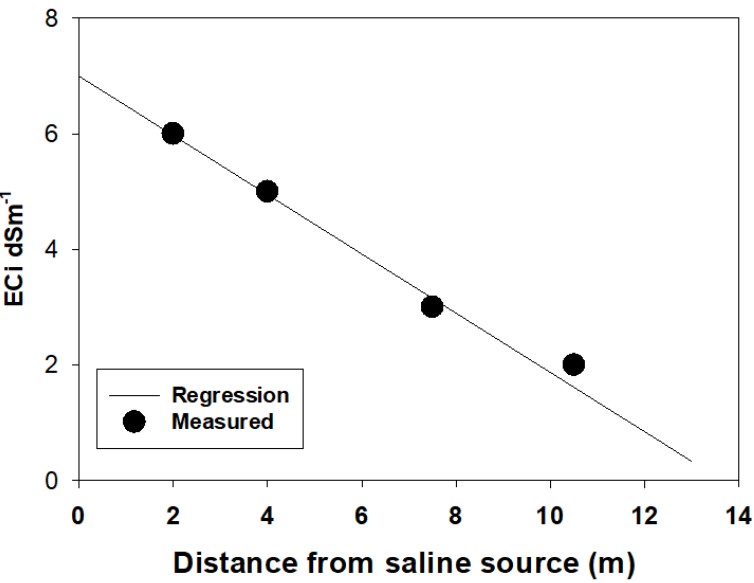

**Figure 1.** Changes in irrigation water salinity (ECi) as a function of distance from the saline source line: Regression Equation (6).

The experiment was a modification of the single line source concept of Hanks et al. [31]. Original salinity was 7.0 dSm$^{-1}$. Samples of water applied to the plot were collected during each irrigation by funnels attached to collection bottles. Total water amount collectd was 202 mm and the cristiansen uniformity was 0.95. Ec decreased linearly awaiy from the saline line (Figure 1).

$$ECi = 7 - 0.5X; r^2 = 0.99 \quad\quad (6)$$

The ECi is reduced due to its dilution with water from fresh line that is located opposite to saline line source.

Christiansen [34] water distribution coefficient was always above 90%, with an average of 94%, showing its precise distribution. Two sprinkler lines were used for irrigation to create salinity gradients from 1 to 8 dS m$^{-1}$ for corn and from 1 to 11 dS m$^{-1}$ for lettuce. Soil water content was monitored gravimetrically to a depth of 0.6 m below the surface.

The gravimetric measurement was complemented by using a Class A evaporation pan [35–37]. Irrigation was done every day, maintaining soil water content at field capacity, being, therefore, on this case, relative soil water availability affected only by salinity and not by soil water content. The control of soil water content was based upon the direct determination of the moisture content and dry weight of the material in the oven at 105 °C until constant weight or with a neutron probe.

The extraction of soil solution was done recurring to suction cups [38–40]. Soil water properties are given in Table 1. Seedbed and basic fertilization were made according to regional conventional agro-techniques. Harvest of sweet corn and lettuce was made about 150 and 35 days respectively. Differences between treatments were analyzed by one-away ANOVA; treatments means were compared by Duncan's multiple-range tests (DMRT) at 95%.

**Table 1.** Soil properties.

| Crop | Soil Texture | Soil pH | Soil Depth (m) | $\theta_{fc}$ m$^3$ m$^{-3}$ | $\theta_{wp}$ m$^3$ m$^{-3}$ | ASW |
|---|---|---|---|---|---|---|
| Sweet Corn | Silty clay loam (loess typic haploxeralf) | 7.6 | 0.0–0.20 | 0.26 | 0.11 | 0.15 |
| Lettuce | Loamy sand (typic palexeralf) | 6.3 | 0.0–0.50 | 0.12 | 0.04 | 0.08 |

## 4. Results and Discussions

In Figures 2 and 3 three models are used: The ASW model (Figure 2), MH and the vGG model (Figure 3). MH model or the threshold model is given by Equation (4) and the vGG model is given in Equation (5).

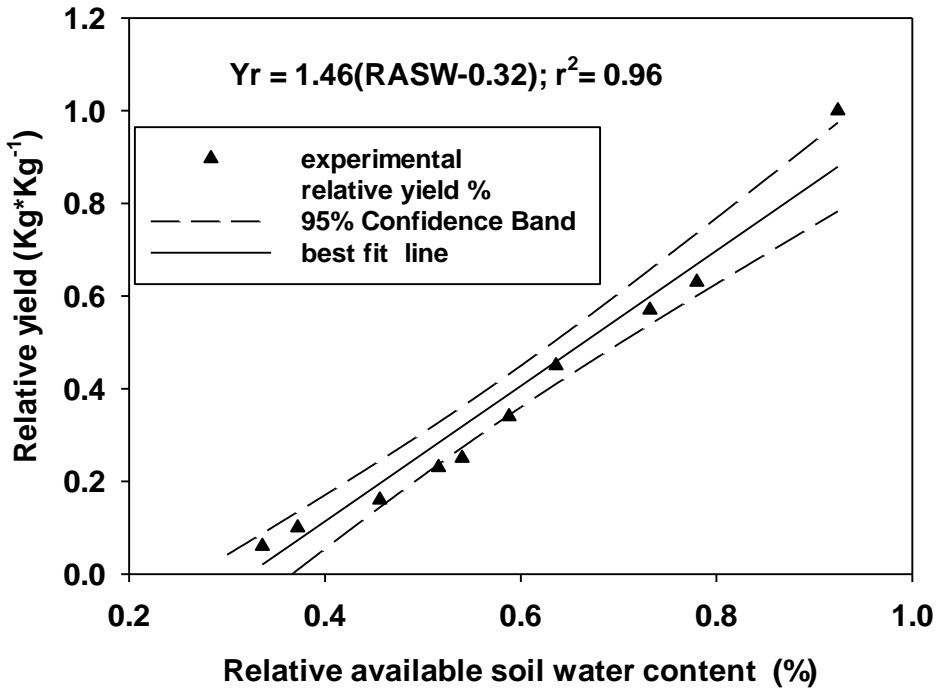

**Figure 2.** Effect of salinity expressed as relative available soil water content (RASW) on relative yield of lettuce.

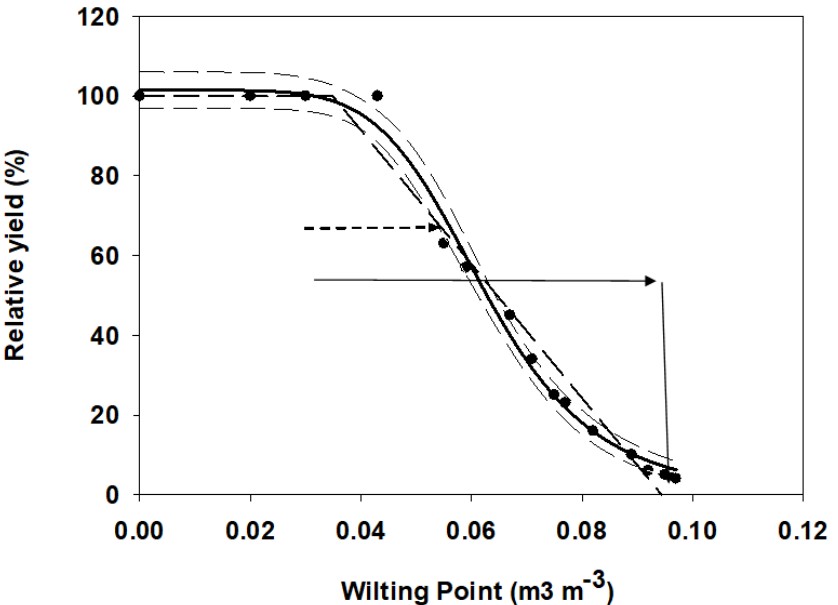

**Figure 3.** Relative yield as a function of increased wilting point affected due to increased salinity. The horizontal dashed arrow points at MH model and the solid lines pointing at the value of WP when relative yield = 50%.

The old Veihemeyer-Hendrickson [24] assumption suggests that with a constant field capacity available soil water depends on the wilting point. The higher the wilting point the lower is water availability and vice versa.

The proposed model emphasized the effect of salinity on the wilting point and modified $EC_e$ of known salinity models (MH and vGG) to ASW on one hand, and to the wilting point on the other. Therefore, crop yield response to availability of saline water was modified and tested in two graphs the first one (Figure 2) shows yield response to relative available soil water (RASW) and the other is a complementary graph that displays crop yield response to wilting point (Figure 3).

The two graphs show that available soil water content is reduced due to the presence of salts in the soil solution. Mathematically, this can be described by a modified linear regression equation that describe an inverted MH model by Figure 2.

As in Equation (1), the lower the wilting point, the higher is soil water availability in the abscissa that is associated with increased crop yield in the ordinate. Experimentally, it is shown in Figure 2.

The best fit analysis demonstrates the mathematical advantage of vGG in Figure 3. As predicted by the best fit parameters of the experimental points vGG comply properly to Equation (5). The central solid line of vGG fit the experimental points ($r^2 = 0.99$). The heavy dashed vertical arrow marks the $\theta_{wp50} \approx 0.0625$ at $Y_{r50}$. The: 95% confidence band is marked by two dashed lines that bordered most experimental points. Straight heavy dashed line is the modified MH salinity threshold model of the same experimental dotes ($r^2 = 0.9$). At Yr = 100 this straight line is intercepted by a horizontal arrow to mark the threshold salinity breaking point.

Contrary to the continuous property of vGG model, the upper limit of MH ($Y_{r0} \approx 100$) is somewhat misleading since the statistical analysis of the correlation between the experimental points (dotted symbols in Figure 3) included the values of $Y_r$ that stretch from $\theta_{wp} = 0$ to 0.03 were not measured but added manually to mark the threshold breaking point, which is highlighted by the horizontal coarse dash arrow in Figure 3. However, the MH straight line fits the experimental points from about $\theta_{wp} \approx 0.05$ to 0.1.

Regression line for vGG is Equation (7) and the MH regression line is Equation (7)

$$Y_r(vGG) = 100 / \left[1 + \left(\theta_{wp}/0.0625\right)^{6.2}\right]; r^2 = 0.99$$

$$Y_r(MH) = \begin{cases} 100 - 4.3 * (WP - 0.035), & WP < 0035 \\ 100, & WP < 0035 \end{cases} ; r^2 = 0.9x \tag{7}$$

Mathematically, applying the nonlinear least square inversion program to the lettuce data resulted in an excellent fit of Equation (5) to the experimental data. In particular, the tailing part at the higher wilting point is now described much better than in the MH modified model.

Detailed conventional relationships of $Y_r$ vs. EC vs. are summarized here in Table 2.

**Table 2.** Salt tolerance of corn and lettuce expressed as electrical conductivity at the threshold when the slope intercepted $Y_r = 100$ *.

| Crop | EC Threshold | Slope%/dS m$^{-1}$ | r$^2$ | (Rating) |
|---|---|---|---|---|
| Sweet corn | 1.1 (1.7) | −7.45 (12) | 0.92 | Moderately tolerant |
| Lettuce | 1.1 (1.3–1.7) | −8.3 (12) | 0.88 | Moderately tolerant |

* In Parenthesis the data extracted from FAO 56 [25].

Table 2 reveals that both threshold and slope of salinity data published in the FAO 56 book [41] were higher than the values measured in our experiment. That is, lettuce and corn of "FAO 56 experiments" did not lose yield until EC was about 1.7 dS m$^{-1}$ while this experiment started to lose yield at EC of about 1.1 dS m$^{-1}$. On the other hand, the yield of "FAO 56 experiments" declined at a rate of 12% per 1 dS m$^{-1}$ while in this experiment it was only about 8% per dS m$^{-1}$ much slower than the crops in "FAO 29 experiments [42]. Thus, it can be shown that using the conventional MH model, at about 3 dSm$^{-1}$ the yield

of the two experiments is equalized at 80% of its maximum, and beyond this the yield of "FAO 56 experiments" is lower than that of this experiment.

Relationship between total simulated and observed relative sweet corn and lettuce relative yields is presented in Figure 4. It can be seen that the slope is very close to 1, the intercept is quite small, and the coefficient of determination $r^2$ (0.97) is very high. It shows that the regression is highly significant, and, therefore, the predicting ability of this approach is excellent and highly capable of describing the relative yield response to the relative availability of soil water as influenced by salinity RASW (s).

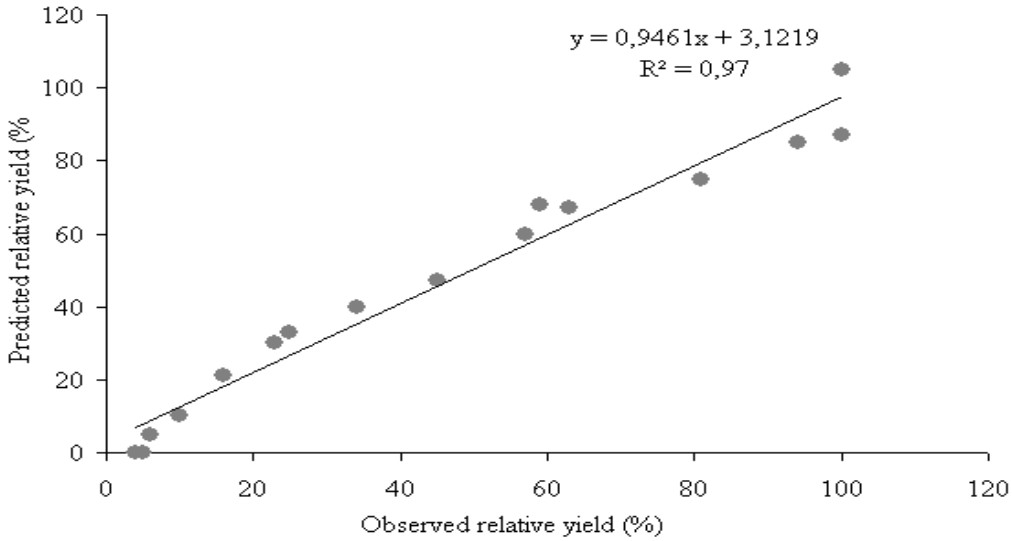

**Figure 4.** Relationship between total simulated and observed relative Sweet Corn and lettuce relative yields.

## 5. Model Validation

Generally model validation is an independent test of its generality. Usually a model is developed on relatively limited number of data under specific experimental conditions. Therefore, its validation on larger amount of data under varied conditions is a must.

Specifically, the objectives of this paragraph are: (A) Validate the theoretical analysis of the proposed model which is based on soil water content considerations (Equation (2)) by comparing it to another widely used theoretical model [43] based on soil physics considerations; and (B) Validate experimental result (Equation (5) and Figure 3) collected from several experiments that were conducted on Golf courses.

Knowing that crop water deficiency or availability are strongly related to dilution of salts in the soil solution we tested and ensured that independent experimental results support the results of model calibration and logic.

Bermuda grass grown on golf courses was used for validation on sandy and sandy loam soils irrigated by water of various salinities. It was selected to this study because the Bermuda grass is the most used species in the fairways of the Mediterranean golf courses.

Validation of Equations (1) and (2) by SPAW (soil plant atmosphere water) model is shown in Table 3 for arbitrary salinity data.

**Table 3.** SPAW calculations of percent volumetric soil water content at wilting point (WP) and available soil water (ASW) as a function of increased arbitrarily electrical conductivity (EC) for sand and clay.

| Soil Texture | SAND Typic Torrisament 90% Sand 4% Clay | | CLAY FLUVIOSOL-Thionic Clay 50% Sand 40% | | |
| --- | --- | --- | --- | --- | --- |
| EC dS/m | %WP | %ASW | EC dS/m | %WP | %ASW |
| 0 | 1.6 | 7 | 0 | 29.7 | 13 |
| 1 | 2.1 | 5 | 1 | 29.9 | 13 |
| 2 | 2.9 | 4 | 2 | 30 | 13 |
| 3 | 3.8 | 3 | 3 | 30.1 | 13 |
| 4 | 4.8 | 2 | 4 | 30.3 | 13 |
| 5 | 5.8 | 1 | 5 | 30.4 | 13 |
| 10 | 11.5 | 0 | 10 | 31.2 | 12 |
| 15 | 17.2 | 0 | 15 | 32.3 | 11 |

SPAW model is a deterministic simulation of soil water tension, hydraulic and electrical conductivities and water holding capability all based on the soil texture.

From Table 3 it can be seen that as expected by Equation (2) increased salinity was also associated with increased WP and reduction in ASW (Equation (1)). Equations (1) and (2) are based on soil water balance considerations. The same directions are identified by SPAW that is based on soils physical properties. The reduction of ASW is strongly affected by soil salinity. In sandy soil ASW does not have any threshold value and its reduction starts consistently from Ec = 0 to Ec = 10 when ASW = 0. In clay there is a long threshold of ASW at 13% and insignificant reduction starts at high Ec = 10 in which most cultivated crops can't survive.

The second validation step was based on data interpretation from published experiments that were conducted on golf courses [44,45]. Point source experimental design that is described in details in the materials and method section was used in this experiment (Figure 1). The gradual change of Bermuda grass production was linked to crop coefficient that varied from low K = 0.1 to high yield with K = 1.6. The differences in water application were expressed by crop coefficients (k 0.1 – k1.6) and converted to soil water contents. Similarly, salinity of the irrigation water (ECi) was gradually changed with distance from the saline water source and thus, its relation to ASW could be obtained iteratively by SWAP and is given by Equation (8).

$$Sandy \quad soil \ ASW = \begin{cases} -1.14 * ECi + 6.5; & ECi < 5 \\ 0 & ECi > 5 \end{cases} \quad r^2 = 0.97$$

$$Clay \quad soil \ ASW = \begin{cases} -0.26 * ECi + 14.5; & ECi > 5 \\ 13 & ECi < 5 \end{cases} \quad r^2 = 0.97 \tag{8}$$

Equation (8) convert ECi data to available soil water (ASW) from which the modified form can calculate WP using SWAP. The process is displayed by a flowchart (Figure 5).

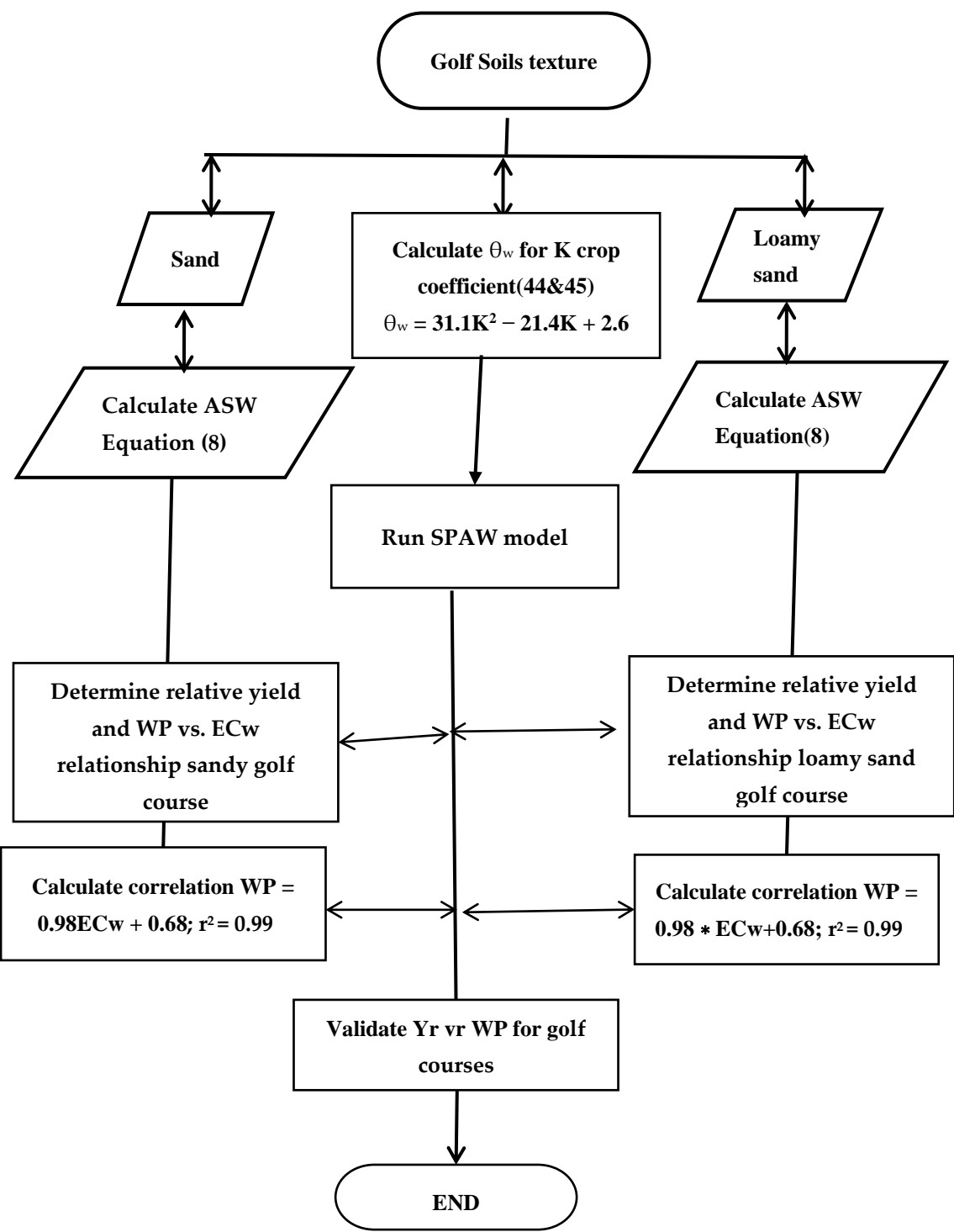

**Figure 5.** Flowchart describing independent validation of Equations (1) and (2) by processing data collected from experiments on golf courses.

The relative yield ($Y_r$) is a function of crop coefficient (K) that in itself related to ASW. The higher was ECi the lower was ASW and hence $Y_r$. For large ECi this effect is more pronounced. The reliability of Equation (8) has been evaluated by Corwin and Lesch (46) who concluded that it is reliable except under extremely dry soil conditions when $\theta_w \to 0$ and ECi $\to \infty$. Theoretically, $EC_i$ is the best index of soil salinity because this is the salinity actually experienced by the plant root. Notice that soil extract ($EC_e$) has been the standard measure of salinity used in all salt-tolerance plant studies. Once $EC_i$ is known, wilting

point can be depicted from the solution of SPAW as demonstrated in Table 3, the flowchart in Figure 5, and Figure 3 that plot of relative yield vs. wilting point.

For the validation of Equations (1) and (2) on golf courses two soils were used. The first one "Fluviosol-thionic", according to the universal soil classification. The texture of this soil is sand and it is composed of 96%; sand, 1% silt and 3% clay. The second is Arenossol Haplic containing 80.7% sand; 12.4% silt and 6.9% clay. Its texture is classified as loam sandy soil. The initial conditions to calculate $\theta_{fc}$ and $\theta_{wp}$ by SPAW model were the percentage of sand, clay and silt for $EC_e = EC_w = 0$. Then $EC_w$ was calculated according to Equation (6). The new $EC_w$ that was obtained from Equation (6) was applied to SPAW model for a new run with the new salinity value new from which a new wilting point value was displayed and presented versus the measured relative yield and increased the salinity input until the suitable WP was obtained. Notice due to increase in osmotic potential only wilting point is increased while field capacity and saturation are not affected by osmotic potential and are remaining constant.

Results of relative yield vs. wilting point are displayed in Figure 6.

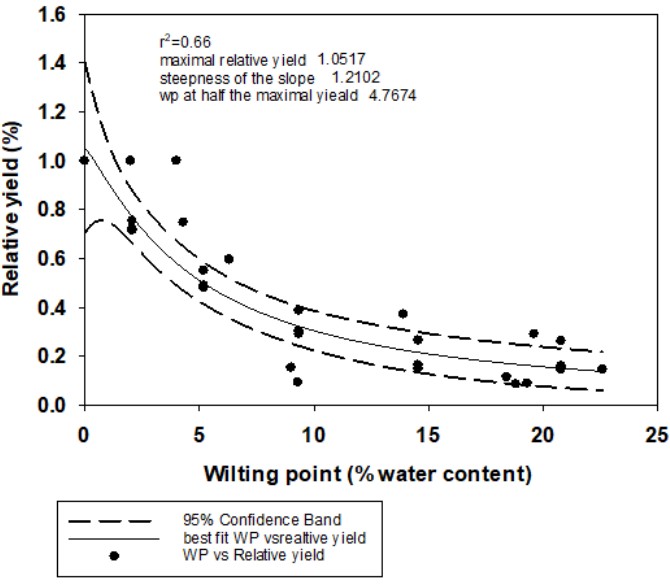

**Figure 6.** Observed data interpreted from experiments on golf courses for Bermuda grass and the 95% confidence band of the best fit.

In Figure 6 the response function doesn't show a typical threshold plateau nor shape though the best fit equation and its parameters provided the expected S response function.

In spite of the missing threshold the data in Figure 5 presents a nonlinear least square inversion program with wilting point on the abscissa that resulted in an acceptable fit ($r^2 = 0.66$) of Bermuda grass yield. In particular, the tailing part at the higher wilting point $\theta$ is now described properly and validate the theory. At low wilting $\theta$ the 95% confidence band is wider than that of the tail and it bends toward the ordinate. The inverse procedure resulted in a uniquely defined curve, independent of the initial parameter estimates with the steepness of the slope the only unknown. This and Table 3 that displays the effect of osmotic water potential on total salinity and its associated increase of wilting point while reducing ASW and yield.

## 6. Conclusions

This article describes a simple modeling approach to calculate crop response to salinity. The model is based on water availability that is a function of salinity and was tested in Israel and Portugal using the single sprinkler point source approach. Independent validation was taken from experiments that were conducted on golf courses using similar experimental methods. Validation was complemented by a theoretical deterministic model that combines

soil physics and soil salinity [46]. The results indicated that MH or vGG models can also be expressed by two versions of water availability and water deficiency due to salts dilution in the soil solution. The generality of the limited observations (only lettuce and corn) was strengthened by independent validation on golf courses and therefore allowed to draw general conclusions.

In conclusion, it can be said that MH work on crop salt tolerance was validated by a set of soil water balance measurements rather than $EC_e$ measurements. The newly proposed method to estimate relative yield vs. $EC_e$ is combining soil water balance ($\theta$) and salinity of irrigation water ($EC_e$) and it is easier to obtain than soil salinity ($EC_w$). It assumes minimum yield due to high WP and its associated low ASW. In relation to ASW the threshold indicating water deficiency is a mirror image of MH model.

Comparing Figure 2 with Figure 3 it can be seen that RASW is inversely related to MH model. Both have a threshold value but the first part of MH model is horizontal along maximum yield and the second is sloping downward describing yield reduction as salinity increase while the first part of RASW is also horizontal but it describes no yield and the second part is sloping upward describing yield increase with increase as RASW increase. The breakthrough points of "availability threshold" can be calculated from the regression equation in Figure 3 (RAWS = 0.53/1.54 = 0.34). This study shows that vGG model given by Equation (5) leads to a better description of experimental data than the MH model. It contains fewer unknowns and may be used to more accurately analyze tolerance to water deficiency data sets. The presence of a unique dimensionless tolerance to water deficit vs. yield response model may point to the increase in soil water tension governing the yield response to salinity, and perhaps to osmotic, water stress and other yield-limiting factors. The difficulty in this perspective is that this research does not meet the mechanistic variables defining the MH and vGG models. Experimental evidence and mechanistic derivation of the experiment may reopen away to increase yield.

**Author Contributions:** Conceptualization, J.B.-A., G.B. and J.B.; methodology, G.B., J.B. and J.B.-A.; software, J.B.-A.; validation, T.P.; writing, J.B.-A. All authors have read and agreed to the published version of the manuscript.

**Funding:** The Fundação para a Ciência e Tecnologia Financed this Paper under grant PTDC/GES-URB/131928/2017. University of Algave Portugal and Ben-Gurion University Israel.

**Institutional Review Board Statement:** Not applicable.

**Informed Consent Statement:** Not applicable.

**Data Availability Statement:** Raw data is available on request from the corresponding author.

**Conflicts of Interest:** The authors declare no conflict of interest.

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
