# Peer review of "Crop Response to Combined Availability of Soil Water and Its Salinity Level: Theory, Experiments and Validation on Golf Courses"

_agronomy, doi:10.3390/agronomy11102012_

Round 1

Reviewer 1 Report

Authors have presented a study of different model describing the effect of salinity through soil water availability on crop production. The topic of the manuscript is interesting, since salinity is an important problem affecting crops in many regions of the world. Following, I have included some comments to improve the manuscript:

  • I suggest to the authors to add a new section detailing state of the art. In this section, authors have to describe better the relevant related work in which explain the different predicting models used, advantage, contribution in yield prediction.
  • The authors should clearly indicate their novelty in this study. Introduction, objective and theory presented are little confusing, improved it.
  • In material and methods, in the part of the cultivar used, authors must add some explication concerning the two species used, if they are sensible o tolerant to salinity, ext. Add some characteristics of the two cultivars studied.
  • Figure 1. Effect of salinity expressed as available soil water content (ASW) on relative yield of lettuce. Why authors do not presented the same figure for sweet grain corn.
  • The results and discussion are very short, improved it. 
  • Relation between real and predicted yield presented in Fig 3 is little confusing. Authors present the observed and predicted for the two species together. Why they do not presented this Fig for each crop.
  • The presentation of the results is poor and confusing.
  • The authors must firstly explain better the chosen model in this study in the section of material and methods. In the results apply the model to each crop; compare better the performance observations with the prediction. All this sections must be improved in the manuscript.
  • Finally, the topic of this manuscript is interesting, but the text does not reflect it. The Manuscript must be improved. Authors must restructure and add more details and explications to their results and discussion to improve the text. They should clarify their objectives and conclusions. Finally, they must improve their results (better presentation of the Anova, more details for the results, and better explanation of the figures).

Author Response

Dear Reviewer,

Thank you for your time and attention.

On behalf of all the authors, I apologize to you.

We were unable to respond in the format you requested because two of the authors were ill, but the article was completely reworked based on your recommendations.

Best regards,

Gulom Bekmirzaev

Reviewer 2 Report

Dear Authors, I am sorry to express my disappointment with your revisions. I was expecting a point by point response to each of the major comment and also some reaction to the minor comments listed directly in manuscript. Although your revised manuscript contains new and current references (my major comment no.2), I did not find almost any reaction to my other comments. That is why I cannot recommend the manuscript for publication unless the revisions are completed and the manuscript is significantly improved.  

Kind regards, your reviewer

Author Response

Dear Reviewer,

Thank you for your time and attention.

On behalf of all the authors, I apologize to you. We were unable to respond in the format you requested because two of the authors were ill, but the manuscript was completely reworked based on your recommendations.

Best regards,

Gulom Bekmirzaev

Reviewer 3 Report

The present form is suitable for consideration.

Author Response

Dear Reviewer,

Thank you for time and attention.

Best regards,

Gulom Bekmirzaev

Round 2

Reviewer 1 Report

I consider that the authors have made the modifications that had been proposed to them properly, so the article is ready for publication.

Reviewer 2 Report

Dear Authors,

Thank you for your patience when waiting for my response. It was not an easy task for me to make the final decision stating that I do recommend to publish the manuscript, but only after some revisions are made. I fully understand your reasoning and I can imagine how difficult it was to respond with two of the co-authors being ill. However, the original manuscript was reworked, but some of the parts requiring revision were not revised. The main change in the manuscript is based on adding a new part dealing with the validation of the newly developed approach. This new part improved the content and increased the impact of the manuscript, no doubt about that. I also think, that a great amount of work is hidden behind the manuscript, but its presentation is rather poor.

In my original review, I had three major points out of which two were not addressed or answered or commented. Major comment 1 - I still think that adding few lines dealing with importance of the manuscript topic and putting it into the context is desirable. Major comment 2- Addressed – newer references were added. Major comment 3 – Not sufficiently addressed – I do not think that adding references instead of mentioning the details of the methodology is suitable. The soils being used in the study should be classified according to some system (WRB, USDA-NRCS, FAO-UNESCO-ISRIC). No weather or climatic conditions are mentioned for the study-sites. The Materials and methods section needs to be elaborated into more details.

In addition to that, some minor mistakes still remained in the manuscript.

You have added the “B)” objective into the “Abstract”, but you did not update the objectives of the study within the “Introduction” section. Generally, the titles of the tables and figures should be descriptive to allow the individual figure or table to stand alone (explaining also presented symbols). Figure 2 – y axis description “Relative yield” instead of “Relative yielg”, decimal “,” instead of decimal “.” in Figure 3, inconsistencies such as use of both  “R2” and “r2”, use of small or capital letters such as Sweet Corn – Sweet corn, Confidence Band, wp – WP for wilting point and others.

In order to conclude my review, you have collected a great data, but its processing and presentation require and deserve some more time and work.

Kind regards,

Your reviewer.  

This manuscript is a resubmission of an earlier submission. The following is a list of the peer review reports and author responses from that submission.

Round 1

Reviewer 1 Report

Authors have presented a study of different model describing the effect of salinity through soil water availability on crop production. The topic of the manuscript is interesting, since salinity is an important problem affecting crops in many regions of the world. However, the manuscript present serious problem. The presentation of results and discussion of the manuscript must be improved. Following, I have included some comments to improve the manuscript:

  • Rewrite objectives. They are poorly presented at the end of the introduction.
  • I suggest to the authors to add a new section detailing state of the art. In this section, authors have to describe better the relevant related work in which explain the different predicting models used, advantage, contribution in yield prediction.
  • The authors should clearly indicate their novelty in this study. Introduction, objective and theory presented are little confusing, improved it.
  • In material and methods, in the part of the cultivar used, authors must add some explication concerning the two species used, if they are sensible o tolerant to salinity, ext. Add some characteristics of the two cultivars studied.
  • In the theory part, the authors have presented the models used in the bibliography and the changes that they contribute in this study, but this section lacks more organization, it still needs to explain the entire proposed model by authors with details.
  • Authors cited in theory: In Figs. 1 and 2 (figures of the results) three models are used: The ASW model (Fig.1), MH and the vGG model (Fig 2). In the theory, the authors already mention the figures of the results. Authors must explain very well the three models of this study regardless of the results.
  • Figure 1. Effect of salinity expressed as available soil water content (ASW) on relative yield of lettuce. Why authors do not presented the same figure for sweet grain corn.
  • The results and discussion are very short, improved it. Separate this section into subdivisions, it is in a single block of text, and this make it less readable.
  • Relation between real and predicted yield presented in Fig 3 is little confusing. Authors present the observed and predicted for the two species together. Why they do not presented this Fig for each crop.
  • The presentation of the results is poor and confusing.
  • The authors must firstly explain better the chosen model in this study in the section of material and methods. In the results apply the model to each crop; compare better the performance observations with the prediction. All this sections must be improved in the manuscript.
  • Finally, the topic of this manuscript is interesting, but the text does not reflect it. The Manuscript must be improved. Authors must restructure and add more details and explications to their results and discussion to improve the text. They should clarify their objectives and conclusions. Finally, they must improve their results (better presentation of the Anova, more details for the results, and better explanation of the figures).

Author Response

Dear Reviewer,
The manuscript has been re-edited according to your review and comments.

Best regards,
Gulom Bekmirzaev

Reviewer 2 Report

Dear authors,

Please see my review report with my comments and reasoning which has led me to the overall recommendation of accepting the manuscript only after major revision.

Your manuscript dealing with the combined effect of salinity and soil water availability on crop yield is interesting and in my opinion also very actual and suitable for „Agronomy“ journal. The decrease of available water for plants due to increase of soil salinity is a well-known effect; however, finding of a simple empirical model to predict crop yield on the basis of relative availability of soil water affected by salinity is an interesting approach. High crop yield obtained by a sustainable way is one of the today’s priorities. Naturally, general statements cannot be derived if the models were tested only on two crop types. On the other hand, it looks like a promising start. Although I like the way you present your study (it is relatively short, easy to follow with clearly defined objectives), I have three major points, which in my opinion needs to be addressed prior the publication of the manuscript. Two of them are dealing with the background literature and one with the missing details in “Materials and Methods” section.

MAJOR COMMENTS

  1. In my opinion, the “Introduction” section should start with some lines describing the importance of the present study with the nowadays agricultural practice (i.e. saline water uses for irrigation, increase of salinity levels of soils being irrigated or increase need of irrigation due to droughts).
  2. Based on the title, I expected more detailed description of the theories and different experiments. It is ok to start with the early papers published in 1931 and 1942, but since than a lot could have been revealed.  In your manuscript you utilized only 6 papers which are younger than 20 years, out of which 4 are younger than 5 years. In my opinion, more recent literature should be used.
  3. I really miss some details about the experimental sites. Experimental fields localization and description, soil classification, climatic and weather conditions, frequency of soil solution extraction. The details might be available within the studies presented in ref. the [20] and [21]. It might be interesting to see the daily amounts of water used for irrigation for each particular field with respect to weather conditions and/or crop transpiration.

MINOR COMMENTS

Minor comments are written directly into the .pdf file.

Kind regards,

Your anonymous reviewer

Author Response

(The authors gave the same response as above.)

Reviewer 3 Report

Manuscript Agronomy – 1104128

A study on crop response to combined availability of soil water and its salinity level: theory and experiments

Reviewer’s comments

This is a very interesting article. However, I think that the authors should pay attention to the following aspects and introduce corresponding minor revisions:

1 – Clearer definition is required for basic terms and concepts as ASW, SWA, RASW (pg. 2/8).

2 – The formulation of the models - MH (threshold), eq.(4) and Fig. 1; vGG (continuity), eq.(5) and Fig. 2 – could be more clearly described. In fact, at a first reading, the presentations seem to refer to a single moment, with the soil at field capacity Ó¨fc and wilting point Ó¨wp just modified by the actual value corresponding to the water salinity. Hence, eqs. (4) and (5) would be point relations, as well as Yr. However, the reader may be expecting yield to be calculated as it is in fact, an integral through the all growing season, the relations varying continuously. The reader should be advised of the abstraction for a given growing phase, with varying salinity in the water and the corresponding Ó¨wp, which becomes a variable. The experiments should deserve a similar descriptive care.

3 – Last paragraph on pg. 3/8, after eq. (5): “Having the yield changing with Ó¨wp, and Yo maximal yield, the ratio Y/Yo = 100”. This period is hard to understand, as it does not seem to correspond to eq. (5).

4 – Pg. 4/8: In 3. Materials & methods, “The gravimetric measurement was complemented by using a class A evaporation pan”. What for? How and when are the evaporation data used?

5 – Table 1 (soil properties): 0.2m depth for maize (silty clay loam, in Israel). Please confirm this soil depth.

6 – Figure 1: the xx axis is labelled %, but numbers show decimal form.

7 – Figure 2: the heavy dashed vertical arrow does not seem to be well placed under Ó¨wp50.

8 – Table 2: why are slopes in these experiments marked negative (-7.45 and -8.3)?

Author Response

(The authors gave the same response as above.)
